# Genome-Wide Identification and Expression Analysis of *GASA* Genes in *Hevea brasiliensis* Reveals Their Involvement in Response to Cold Stress

**DOI:** 10.3390/ijms26073454

**Published:** 2025-04-07

**Authors:** Yuying Cong, Yanshi Hu, Zhi Deng, Wenguan Wu, Tingkai Wu, Yanhong Zhao, Zewei An

**Affiliations:** 1National Key Laboratory of Tropical Crop Breeding, Rubber Research Institute, Chinese Academy of Tropical Agricultural Sciences, Haikou 571101, China; shixiaohe178@163.com (Y.C.); mfcjason@163.com (Y.H.); zizip@163.com (Z.D.); wenguanwu88@163.com (W.W.); wtksicau@163.com (T.W.); 2School of Agriculture, Ludong University, Yantai 264025, China; zyhbob@163.com; 3Sanya Research Institute, Chinese Academy of Tropical Agricultural Sciences, Sanya 572000, China

**Keywords:** *Hevea brasiliensis*, *GASAs*, expression profile, cold stress

## Abstract

The Gibberellic Acid Stimulated in *Arabidopsis (GASA*) gene family is regulated by gibberellins and plays a crucial role in regulating plant growth and development. Based on rubber tree genome data, 18 *HbGASA* genes, designated *HbGASA1* to *HbGASA18*, were identified in *Hevea brasiliensis*. Comprehensive bioinformatics analyses were performed to characterize gene structures, chromosomal distributions, syntenic relationships, protein architectures, phylogenetic evolution, and expression profiles. The expression patterns of *HbGASA* genes under low-temperature stress were further validated by quantitative real-time polymerase chain reactions (qRT-PCR). The results demonstrated that the 18 *HbGASA* genes were unevenly distributed across 10 chromosomes. The encoded proteins ranged from 88 to 253 amino acids in length, and the number of exons varied from 2 to 4. Phylogenetic analysis clustered these genes into three distinct clades. Conserved motif analysis identified 10 conserved motifs, with Motif 1 and Motif 2 being highly conserved across all members. Promoter analysis revealed multiple hormone-responsive and stress-related regulatory *cis*-acting elements. Transcripts of the 18 *HbGASA* genes were detected in various tissues, and significant differences were observed in their expression levels. Under cold stress, qRT-PCR results showed that multiple *HbGASA* genes were significantly up-regulated. This study provides valuable insights into the structure, evolution, and functional diversification of *GASA* genes in the important tropical crop, *H. brasiliensis.*

## 1. Introduction

The *GASA* gene family, which is widely distributed in plants, is a cysteine-rich gene family regulated by gibberellins (GA). It plays important roles in plant growth and development, as well as in response to biotic or abiotic stress [1]. Typically, the GASA protein contains three parts: an N-terminal signal peptide with the characteristics of an extracellular secreted protein, composed of 18 to 29 amino acid residues; a variable hydrophilic segment composed of 7 to 31 polar amino acid residues, in which there are significant variations among different GASA proteins; a C-terminal conserved domain of 60 amino acids, known as GASA domain, containing 12 fully conserved cysteine residues [2]. Generally, a *GASA* gene contains only one GASA domain, in which the deletions or changes in key amino acids can make the gene lose its function [3,4]. Therefore, this domain is considered to be the key region for maintaining the spatial structure and function of GASA proteins.

In plants, *GASAs* were usually strongly expressed in young and vigorously growing tissues and organs, with most members under gibberellins regulation [5,6]. Analysis of *GASAs* sequences showed that the promoter region was enriched with *cis*-acting elements related to development, phytohormones, and stress, indicating their dual roles in growth regulation and stress adaptation. As various plant *GASAs* had been identified, it was demonstrated that *GASAs* played important regulatory roles in plant growth and development, including seed germination [7], flower and fruit development [8], lateral root formation [9], hormone signal transduction [10], and response to biotic or abiotic stress [11,12]. Notably, *AtGASA4* was rapidly up-regulated under short-term heat treatment, and the heat tolerance of *Arabidopsis* plants were enhanced by overexpressing *AtGASA4* [13]. Conversely, *AtGASA5* gene acted as a negative regulator in heat-tolerance by regulating both SA signaling and heat shock-protein accumulation [14]. *SmGASA4* isolated from *Salvia miltiorrhiza*, improved *Arabidopsis* resistance to salt, drought, and paclobutrazol (PBZ) [15]. *AtGASA14* regulated the resistance to abiotic stress by modulating the accumulation of reactive oxygen species [16]. *OsGASR1* enhanced plant tolerance to salt stress via the reactive oxygen species scavenging system in *Oryza sativa* [17]. In *Glycine soja*, *GsGASA1* was involved in inhibiting root growth under chronic cold stress via the accumulation of DELLA proteins [18].

The rubber tree (*Hevea brasiliensis*) is an important tropical cash crop and the primary source of natural rubber all over the world [19]. Natural rubber is an essential raw material, widely used in medical and health care, transportation, aerospace, and other fields [20]. The rubber tree, originating in the high-temperature, high-humidity, and windless Amazon River basin, is a typical tropical rainforest species with weak resistance to low temperature. China’s rubber-planting areas are located between 18° N and 24° N, belonging to non-traditional rubber-planting areas, in which rubber tree often suffered from abiotic stresses like low temperature, seasonal drought, typhoons [21]. Among these environmental constraints, low temperature is the most severe disaster affecting rubber tree growth. Therefore, identification of genes associated with cold stress is very important for cold-stress-resistant molecular breeding of rubber tree.

In the present work, the *GASA* genes of *H. brasiliensis* were genome-wide identified; meanwhile, comprehensive characterization of their structural features, syntenic relationships, and expression patterns of all the *HbGASA* genes were investigated. Furthermore, Expression dynamics of *HbGASA* genes under cold stress were evaluated. Our research provided some clues to elucidate the potential roles of GASA proteins in rubber tree development and stress adaptation, which may contribute to further functional investigations on the molecular mechanisms underlying cold tolerance in *H. brasiliensis*.

## 2. Results

### 2.1. Identification and Characteristics of HbGASA Genes

In rubber tree, a comprehensive exploration of GASA-domain encoding genes was carried out. Using the HMMER tool, a total of eighteen genes, namely *HbGASA1*-*HbGASA18*, were successfully identified. Chromosomal localization analysis revealed that these 18 *HbGASAs* were distributed across 10 chromosomes in a rather dispersed manner (Figure 1). On Chromosome 12, a gene cluster composed of *HbGASA1* and *HbGASA11* was formed. Notably, Chromosome 12 harbored the highest number of *HbGASA* genes, with a count of 4, surpassing the quantity on other chromosomes. Sequence analysis further unveiled detailed characteristics of the corresponding proteins (Table 1). The lengths of the 18 HbGASA proteins spanned from 88 to 253 amino acids. Specifically, *HbGASA18* encoded the largest polypeptide, reaching 25aa, while *HbGASA2* and *HbGASA9* encoded the shortest, with only 88aa. In terms of relative molecular weight, it ranged from 9680.57 to 26546.86 Da. HbGASA18 exhibited the largest molecular weight, and HbGASA2 the smallest. Regarding the isoelectric point, it varied from 7.46 to 9.98. HbGASA5 had the lowest isoelectric point, and HbGASA18 the highest. Significantly, the isoelectric points of all 18 proteins were above 7.0, indicating that they all belonged to the category of alkaline proteins. Analysis of the instability index showed that five proteins had an instability index less than 40, while the other 13 proteins were unstable, with an instability index greater than 40. In terms of hydrophobicity, except for *HbGASA5* and *HbGASA14*, which encoded hydrophobic proteins, the remaining 16 *HbGASAs* encoded hydrophilic proteins. The subcellular localization of a protein is intimately associated with its biological function. By performing in silico predictions using the WoLF PSORT tool, the majority of HbGASA proteins were predicted to localize extracellularly. Notable exceptions were HbGASA9 and HbGASA13, which showed chloroplastic localization signals.

### 2.2. Protein and Gene Structure of HbGASA Genes

The protein sequences of HbGASAs were aligned with those of GASA proteins from *Arabidopsis*, poplar, rice, and wheat. The alignment results revealed that all GASA proteins, with the exception of TaGASA16, were grouped into three subfamilies. Each subfamily encompassed homologous proteins originating from five distinct plant species (Figure 2). Specifically, within rubber tree’s GASA proteins, there were three pairs of homologous genes. HbGASA7-HbGASA8 belonged to subfamily A, while HbGASA13-HbGASA14 and HbGASA17-HbGASA18 were part of subfamily C. Overall, the phylogenetic analysis demonstrated that the GASA proteins of rubber tree had a closer evolutionary relationship with the homologous proteins in poplar than with those in the other plant species examined.

Using the SignalP 5.0 server, analysis of HbGASA protein sequences revealed that all 18 members of this family contained N-terminal signal peptides with exceptionally high confidence scores (97.381–99.993%). These signal peptides were classified as the SP (Sec/SPI) type, strongly indicating their involvement in classical secretory pathways. To thoroughly clarify the characteristics of HbGASA proteins, the full-length HbGASA protein sequences were aligned with GeneDoc 2.7 software. The results demonstrated that all 18 HbGASA proteins exhibited a comparable structure. Each of them contained an N-terminal signal peptide ranging from 18 to 30 amino acids and a highly conserved C-terminal domain (GASA domain). Comprising approximately 60 amino acids, this C- terminal domain harbored 12 cysteine residues (Figure 3 and Figure 4B).

Gene structure analysis unveiled that all 18 *HbGASAs* were split genes, with exons being separated by introns. Each gene displayed structural disparities to different extents. The coding regions of *HbGASAs* encompassed 1 to 3 introns (Figure 4A). A total of 10 conserved motifs were detected in the 18 HbGASA proteins (Figure 4C). Notably, all HbGASA proteins harbored Motif 1 and Motif 2, which together form the core GASA domain architecture for biological activity. Moreover, all HbGASA proteins except HbGASA13 contained Motif 3. In contrast, Motif 7 was exclusively present in HbGASA7 and HbGASA8, and Motif 9 was only found in HbGASA17 and HbGASA18, respectively. These observations highlighted both conserved functional cores and lineage-specific adaptations within the *HbGASA* family.

### 2.3. Promoter Region Analysis of HbGASA Genes

Analysis of *cis*-acting elements revealed that the promoters of *HbGASA* genes harbored various elements associated with growth, stress responses, and phytohormone responses (Appendix A). Phytohormone-responsive elements for ethylene, abscisic acid (ABA), methyl jasmonate (MeJA), gibberellic acid (GA), and salicylic acid (SA) were particularly abundant. Additionally, cold-, drought-, and disease-responsive motifs were identified in the majority of *HbGASA* promoters. The promoters of 18 *HbGASA* genes, excluding *HbGASA9*, harbored the STRE element. Remarkably, the promoter of *HbGASA13* exhibited the greatest diversity of *cis*-acting elements. These elements were associated with environmental signals, including light, low temperature, and disease stress, as well as with hormone responses, such as those to GA, MeJA, and auxin. This finding suggests that *HbGASA13* could play a crucial role in plant growth and the response to abiotic stress.

### 2.4. HbGASAs Duplication Analysis

Gene duplication, particularly segmental duplication and tandem duplication, represents the major driving force for gene family expansion. To characterize the homologous evolutionary dynamics of *HbGASAs* in rubber tree and their ancestral lineages, a comprehensive bioinformatics pipeline was implemented. This pipeline integrated chromosomal mapping of *HbGASA* genes, systematic identification of tandem repeat segments, and construction of inter-chromosomal homology networks (Figure 5). Notably, four family members (*HbGASA1*, *HbGASA3*, *HbGASA6*, and *HbGASA11*) displayed no inter-chromosomal homologs, suggesting their origin through ancient single-gene duplication events or horizontal gene transfer. In contrast, the remaining 14 *HbGASAs* formed homologous pairs with genes located on different chromosomes, providing strong evidence for segmental duplication-driven expansion during evolution. Chromosomal synteny analysis further revealed collinear relationships among *HbGASA12* (chromosome 7), *HbGASA9* (chromosome 8), and *HbGASA5* (chromosome 12), indicating these genes likely arose from large-scale genome duplication events that have shaped the genome architecture of rubber tree.

### 2.5. The Tissue-Specific Expression Patterns of HbGASA Genes

Based on publicly available RNA-seq datasets of rubber tree, remarkable differences in the expression levels of the 18 *HbGASAs* were observed across various tissues (Figure 6). To further validate the tissue-specific expression patterns of *HbGASAs,* all *HbGASA* genes were detected by qRT-PCR. The melting curve analysis of amplification products exhibited single sharp peaks, confirming the specificity of target fragments (Appendix A). Among five distinct tissues of rubber tree, viz., bark, leaves, latex, female flowers, and male flowers, the expression of *HbGASA* genes demonstrated tissue-specific characteristics (Figure 7). Specifically, *HbGASA1*, *HbGASA2*, *HbGASA4*, *HbGASA6*, *HbGASA10*, *HbGAS11*, *HbGASA16*, *HbGASA17,* and *HbGASA18* exhibited higher expression levels in male flowers. *HbGASA3*, *HbGASA9*, *HbGASA12,* and *HbGASA14* were more highly expressed in bark. Moreover, *HbGASA5, HbGASA7*, *HbGASA8*, *HbGASA13,* and *HbGASA15* had elevated expression levels in female flowers.

### 2.6. Expression Profiles of HbGASAs Under Cold Stress

Analysis of promoter *cis*-acting elements of *HbGASAs* indicated that the promoter regions harbored cold-responsive elements, suggesting potential transcriptional regulation of *HbGASAs* by low temperature. To investigate cold stress responses, the transcript level of *HbGASAs* was analyzed under cold stress via qRT-PCR. Most of *HbGASAs* displayed a biphasic expression pattern, characterized by initial up-regulation followed by down-regulation (Figure 8). Specifically, *HbGASA2*, *HbGASA3*, *HbGASA7*, *HbGASA10*, *HbGASA13*, *HbGASA15*, *HbGASA16*, *HbGASA17*, and *HbGASA18* responded rapidly to cold stress. Their expression levels reached the highest points merely 2 h after the onset of cold stress. Meanwhile, *HbGASA5, HbGASA9* and *HbGASA14* reached their peak expression levels 6 h post cold stress exposure. Among them, *HbGASA8* demonstrated the most rapid response to cold stress during the low-temperature treatment. Its expression level peaked after only 0.5 h of cold stress exposure. In contrast, *HbGASA4* and *HbGASA6* showed significant down-regulation, indicating negative regulation by cold stress.

## 3. Discussion

The GASA gene family, a plant-specific gene family regulated by gibberellins, has attracted extensive research attention in recent years. In the present study, a total of 18 *HbGASA* genes were identified, which is more than reported in the previous study [22]. This increase can be attributed to the availability of a high-quality reference genome that has been published in recent years [23]. In *Arabidopsis*, the GASA genes were dispersed across all chromosomes. In contrast, in *Hevea brasiliensis*, akin to plants such as rice [24], grapevine [25], and cotton [26], the 18 GASA genes (*HbGASA1*-*HbGASA18*) were unevenly distributed among 18 chromosomes. This non-random distribution might be associated with the evolutionary history of rubber tree and the functional specialization of different chromosomal regions.

Phylogenetic and syntenic analyses of *HbGASA* genes were conducted. The 18 GASA proteins clustered into three subfamilies, consistent with phylogenetic classifications reported in *Malus domestica* [4], *Ricinus communis* [27], *Populus trichocarpa* [28], *Arabidopsis* [29], and *Triticum aestivum* [30]. Previous studies have highlighted that tandem, segmental, and whole-genome duplications (WGD) play critical roles in species evolution [31]. In this study, we systematically analyzed *HbGASA* gene duplication events. Fourteen *HbGASA* genes formed homologous pairs with genes located on distinct chromosomes (Figure 6), suggesting segmental duplication origins. Notably, these duplicated gene pairs were clustered together in the same phylogenetic clade (Figure 4A) and exhibited similar expression patterns under cold stress (Figure 8). These findings corroborate previous observations in *Populus* [32]. For instance, after drought treatment, the expression trends of WGD-derived gene pairs *PeuGASA9/PeuGASA15* and *PeuGASA6/PeuGASA13* were highly consistent, respectively.

The proteins encoded by *HbGASA* genes are all small polypeptides, which is consistent with the findings in *Arabidopsis* [29] and potato [33]. This conservation during evolution implies that the GASA gene family plays fundamental and universal roles in basic physiological processes of plants. Motifs, which are structural aggregates within protein molecules, have the capacity to determine the specific functions of proteins or form specific structural domains within proteins. Common motifs in protein families play pivotal roles in both the functional expression and structural stability of these proteins [4].

In this study, a comprehensive motif analysis was carried out on the 18 members of the GASA protein family in *H.brasiliensis*. The protein architectures of *HbGASAs* provided valuable insights into their potential functions. Notably, the presence of Motif 1 and Motif 2 in all 18 HbGASA proteins, which together constituted the conserved GASA domain, underscored their essential role in maintaining the structure and function of HbGASA proteins. Furthermore, the additional motifs, such as Motif 4 found in HbGASA4, HbGASA7, and HbGASA8, might confer unique functions to these proteins, especially considering their higher expression levels in male flowers. It is thus hypothesized that Motif 4 could be intricately involved in regulating the reproductive development in rubber tree.

The tissue-specific expression patterns of *HbGASA* genes vividly illustrated their remarkable functional diversity. Distinct *HbGASA* genes exhibited preferential expression in particular tissues, including bark, leaves, latex, female flowers, and male flowers. This tissue-specific expression is likely intrinsically linked to the diverse physiological functions of these tissues. For instance, *HbGASA3*, *HbGASA9*, *HbGASA12,* and *HbGASA14, which* were highly expressed in bark, might participate in processes like pathogen defense or the regulation of latex production. In contrast, genes expressed in flowers were probably pivotal for reproductive development. In this study, 14 of 18 *HbGASAs* exhibited higher expression levels in flowers. In apples, buds, and flowers also accumulated high level of *MdGASA* genes, except *MdGASA3* and *MdGASA13/26*, which were more active in fruits [4]. It has been demonstrated that GASA proteins have an impact on flower development and blooming period. *AtGASA4* promotes blooming in *Arabidopsis* [3], but *AtGASA5* represses or delays flowering [8]. The suppression of *AtGASA6* caused late flowering, while over-expression of *AtGASA6* induced early flowering in *Arabidopsis* [34]. Gaining a deep understanding of the regulatory mechanisms governing tissue-specific expression is of utmost importance for accurately deciphering the precise functions of *HbGASA* genes. It will provide key insights into how these genes contribute to the overall growth, development, and survival strategies of rubber tree in different tissue-specific contexts.

In this study, the promoter analysis of *HbGASA* genes uncovered the existence of numerous *cis*-acting elements associated with hormone and stress responses. The prevalence of elements responsive to ethylene, GA, MeJA, ABA, and SA strongly implied that *HbGASA* genes were probably intricately involved in complex hormonal regulatory networks. The identification of cold-related elements in the promoters of most *HbGASA* genes were consistent with the observed expression patterns of multiple *HbGASA* genes under low-temperature stress. All 18 *HbGASA* genes showed varying expression levels and response speeds (Figure 8). This is primarily attributed to promoter activity in stress-responsive gene activation being dependent on both the types and quantities of *cis*-acting elements. The *Cor15a* and *Cor15b* promoters are two homologous promoters of cold-stress-responsive genes. However, in transgenic tomato and tobacco (*Nicotiana tabacum*), the *Cor15a* promoter, which contains two CRT/DRE elements, was significantly more active than the *Cor15b* promoter, which contains one CRT/DRE element [35]. Additionally, *HbGASA* genes have been reported to be regulated by fungal pathogens [22]. Remarkably, the extensive types of *cis*-acting elements harbored in the *HbGASA13* promoter highlighted its critical role in regulating plant growth and abiotic stress responses. However, the molecular mechanisms underlying the interaction between these elements and transcription factors, as well as their precise regulatory effects on gene expression, remain to be fully characterized. Deciphering these regulatory networks is essential for achieving a comprehensive understanding of *HbGASA* gene functions in plant development and stress adaptation.

## 4. Materials and Methods

### 4.1. Plant Materials and Stress Treatment

One-year-old bud-grafted seedlings of rubber clone Reyan73397 were provided by the National Rubber Germplasm Repository of China. To investigate the expression dynamics of *HbGASA* genes under cold stress, seedlings were transferred to a plant growth room at 0 °C, with a 16 h photoperiod and 80 ± 5% relative humidity. The treatment duration was 24 h. Leaves of the plantlets were sampled at 0, 0.5, 2, 4, 6, 8, and 24 h post-treatment initiation. Harvested leaves were immediately frozen in liquid nitrogen and stored at −80 °C for RNA extraction. Three biological replicates were collected for each time point.

### 4.2. Identification and Chromosomal Distribution of HbGASA Genes

Genome sequences, protein sequences, and annotation information of rubber tree were downloaded from the NCBI database (https://www.ncbi.nlm.nih.gov/, accessed on 3 November 2024). The Hidden Markov Model (HMM) profile of the GASA domain (PF02704), retrieved from the Pfam database (http://pfam-legacy.xfam.org/, accessed on 3 November 2024), was employed to search for GASA proteins in rubber tree genome via the HMMER program. To validate the presence of intact GASA domains and extract them from candidate protein sequences, the NCBI Conserved Domain Database (https://www.ncbi.nlm.nih.gov/Structure/cdd/cdd.shtml, accessed on 5 November 2024) and the Simple Modular Architecture Research Tool (SMART; http://smart.embl-heidelberg.de/, accessed on 5 November 2024) were utilized. After eliminating redundant sequences, non-redundant candidate protein containing the conserved GASA domain were selected for downstream analysis. Chromosomal localization of *HbGASA* genes was visualized and mapped using TBtools 2.0.

### 4.3. Physicochemical Properties of HbGASA Proteins

Physicochemical properties of HbGASA proteins, such as the theoretical molecular weight (MW), the isoelectric point (pI), the instability index (II), the aliphatic index, Asp + Glu, Arg + Lys, and the grand average of hydropathicity (GRAVY), were analyzed using Protparam (https://web.expasy.org/protparam/, accessed on 10 November 2024). Secondary structure of HbGASA proteins was predicted via SOPMA (https://npsa-prabi.ibcp.fr/cgi-bin/npsa_automat.pl?page=npsa_sopma.html, accessed on 10 November 2024).

### 4.4. Subcellular Localization Prediction and Signal Peptide Analysis of HbGASA Proteins

The subcellular localization of the GASA family proteins in *H. brasiliensis* was predicted using the online WoLF PSORT tool (https://wolfpsort.hgc.jp/, accessed on 27 February 2025). Protein signal peptides were analyzed using SignalP-5.0 tool (https://services.healthtech.dtu.dk/services/SignalP-5.0/, accessed on 27 February 2025).

### 4.5. Analysis of Conserved Motifs and Gene Structures

Conserved domain analysis was carried out using NCBI’s CD-search tool (https://www.ncbi.nlm.nih.gov/Structure/cdd/wrpsb.cgi, accessed on 5 December 2024). Meanwhile, MEME (http://memesuite.org/tools/meme, accessed on 5 December 2024) was utilized to predict the conserved motifs of HbGASA proteins, and visual outputs in SVG format were exported to facilitate the generation of conserved domain architecture diagrams and motif distribution maps. In addition, the exon-intron analysis of *HbGASA* genes was conducted using the Gene Structure View module in TBtools 2.0.

### 4.6. Analysis of Promoters and Phylogeny of HbGASA Genes

To identify potential regulatory elements, the 2 Kb upstream sequences of the start codon ATG of *HbGASA* genes were extracted from rubber tree genome data and served as promoter sequences. The PlantCARE tool (http://bioinformatics.psb.ugent.be/webtools/plantcare/html/, accessed on 5 November 2024) was employed to predict and analyze the characteristics of *cis*-acting elements in the promoters. Subsequently, TBtools 2.0 was used to create *cis*-acting element analysis maps. For evolutionary analysis, HbGASA protein sequences were aligned with GASA homologs from different species including *Arabidopsis*, poplar, rice, and wheat, which were downloaded from Phytozome (https://phytozome-next.jgi.doe.gov, accessed on 3 November 2024) and the NCBI database (https://www.ncbi.nlm.nih.gov/, accessed on 3 November 2024). Multiple sequence alignment was conducted using the Clustal W algorithm in MEGA 11.0.13 software. A phylogenetic tree was constructed via the neighbor-joining (NJ) method with 1000 bootstrap replicates In MEGA 11.0.13 software. Finally, the phylogenetic tree was generated using the online software Evolview (https://www.evolgenius.info/evolview-v2/#login, accessed on 18 November 2024).

### 4.7. Collinearity Analysis and Visualization of HbGASA Genes

Referring to the experimental method of Xue et al. [36], a script was used to extract the information files required by TBtools 2.0 such as chromosome length, chromosome gene information, and link information between chromosomes based on collinearity and GFF annotation files. These files were then imported into the Advanced Circus plug-in TBtools 2.0 to complete the collinearity analysis, and to draw a collinearity map between chromosomes in rubber tree.

### 4.8. Tissue-Specific Expression Profiles of HbGASA Genes

RNA-seq data for *H.brasiliensis* were obtained from the HeveaDB database (http://hevea.catas.cn, accessed on 20 November 2024). Fragments Per Kilobase of exon per Million reads mapped (FPKM) values of *HbGASA* genes were retrieved and normalized using the DESeq2 pipeline. Based on the normalized FPKM values, tissue-specific expression profiles of *HbGASA* genes in five distinct tissues, namely bark, leaves, latex, female flowers, and male flowers, were acquired. Subsequently, TBtools 2.0 was employed to construct an expression heatmap of *HbGASA* genes. To validate tissue-specific expression patterns, quantitative real- time polymerase chain reactions (qRT-PCR) of *HbGASA* genes was carried out on three biological replicates of each tissue.

### 4.9. RNA Isolation and qRT-PCR Reaction

Total RNA was extracted from different tissues (bark, leaves, latex, female flowers, and male flowers) as well as leaves exposed to various cold stress conditions. The extraction was performed using the RNAprep Pure Plant Plus Kit (TIANGEN, Beijing, China) following the manufacturer’s instructions. The concentration and integrity of extracted RNA were evaluated using the NanoDrop One spectrophotometer (Thermo Fisher Scientific Inc., Waltham, MA, USA) and agarose gel electrophoresis. Subsequently, RNA samples were reverse-transcribed into cDNA with the PrimeScript™ RT reagent Kit with gDNA Eraser (Takara, Dalian, China). The cDNA was diluted five fold. All qRT-PCR assays were conducted using the 2×Q3 SYBR qPCR Master Mix (TOLOBIO, Shanghai, China) according to the manufacturer’s protocol. The reactions were run on a CFX96 Touch Real-Time PCR Detection System (Bio-Rad, Hercules, CA, USA). The qRT-PCR primers, designed with Primer 3 software, were listed in Table 2. *HbYLS8* was selected as the reference gene. Relative expression levels were calculated using the 2^−ΔΔ*Ct*^ method. Each assay included three biological replicates and three technical replicates. One-way ANOVA analysis and graph plotting of the data were performed using GraphPad Prism 10.1.2 software. Multiple comparisons were evaluated based on Tukey’s test to calculate *p*-value.

## 5. Conclusions

A total of eighteen *HbGASA* genes were identified in *H.brasiliensis*, all of which contained a conserved GASA domain, and were phylogenetically classified into three groups. Tissue-specific expression profiling indicated preferential expression of most *HbGASA* genes in bark, leaves, and floral tissues (both male and female flowers), with significantly lower transcript abundance observed in latex. The analysis of promoter *cis*-acting elements revealed that *HbGASA* genes were involved in plant growth, development, and stress responses. Moreover, quantitative real-time PCR (qRT-PCR) analysis suggested that *HbGASA* genes were potentially involved in cold stress responses. Collectively, this study provided a systematic characterization of *HbGASA* genes, offering a critical resource for functional validation and breeding applications in rubber tree.

## Figures and Tables

**Figure 1 ijms-26-03454-f001:**
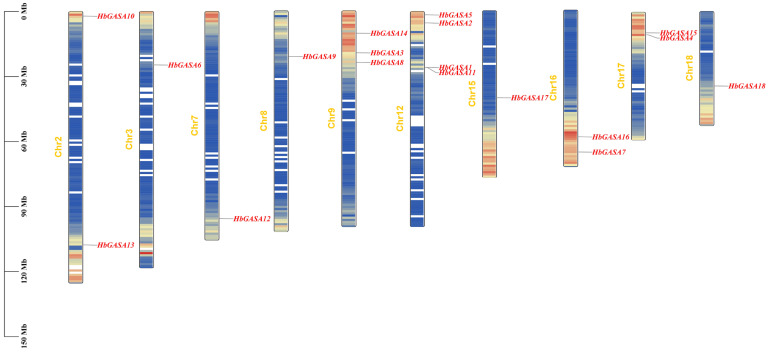
Chromosome location of *HbGASAs* in rubber tree.

**Figure 2 ijms-26-03454-f002:**
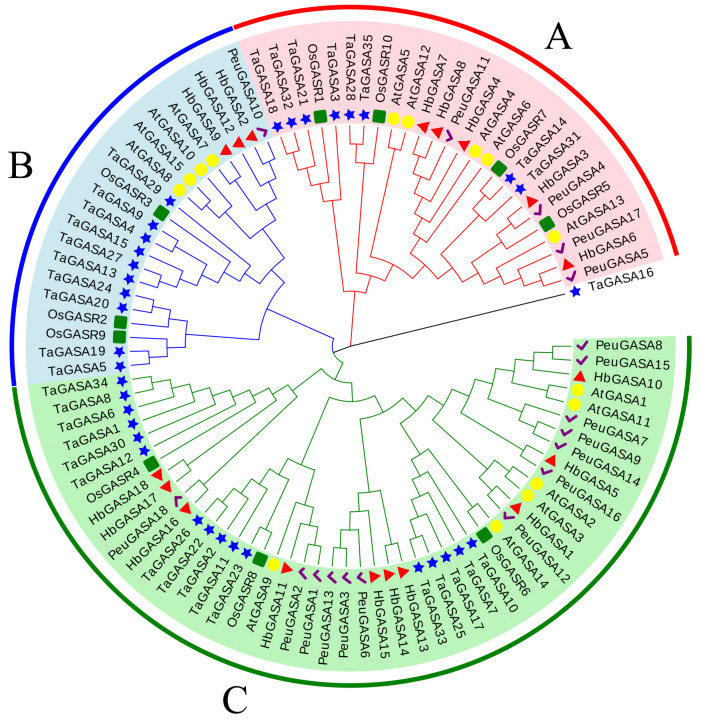
Phylogenetic tree of GASA protein sequences from various plant species. Proteins from five species were indicated in different colors symbols. Red-colored triangles represented rubber tree proteins, green-colored squares represented rice proteins, yellow-colored circles represented *Arabidopsis* proteins, blue-colored pentagons represented wheat proteins, and purple-colored check marks represented poplar proteins. Different colored fan shapes indicated different subfamilies (**A**–**C**).

**Figure 3 ijms-26-03454-f003:**
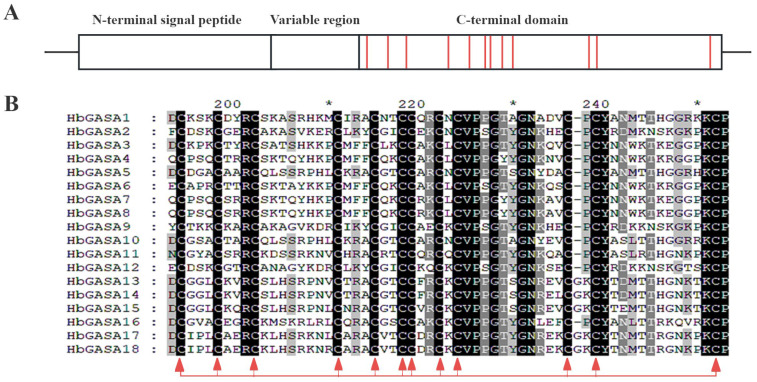
Features of HbGASA proteins. (**A**) Schematic diagram of HbGASA protein structure. The 12 cysteine residues were presented in the red-colored column. (**B**) Multiple sequence alignments of the C-terminal domains of HbGASA proteins. The 12 cysteines residues were marked in red arrows. * represented an interval of 10 aa.

**Figure 4 ijms-26-03454-f004:**
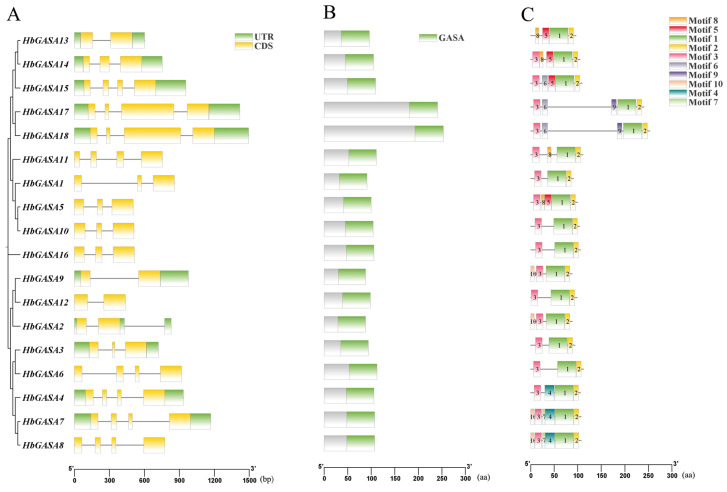
Analysis of *HbGASA* genes and proteins. (**A**) The phylogenetic tree of HbGASA proteins and gene structure; (**B**) the conserved domain; (**C**) the distribution of conserved motifs.

**Figure 5 ijms-26-03454-f005:**
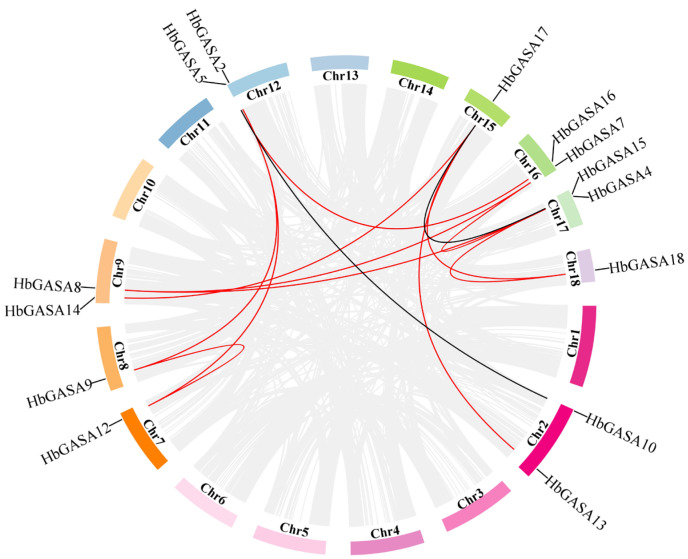
Homologous evolution analysis of *HbGASA* genes in rubber tree.

**Figure 6 ijms-26-03454-f006:**
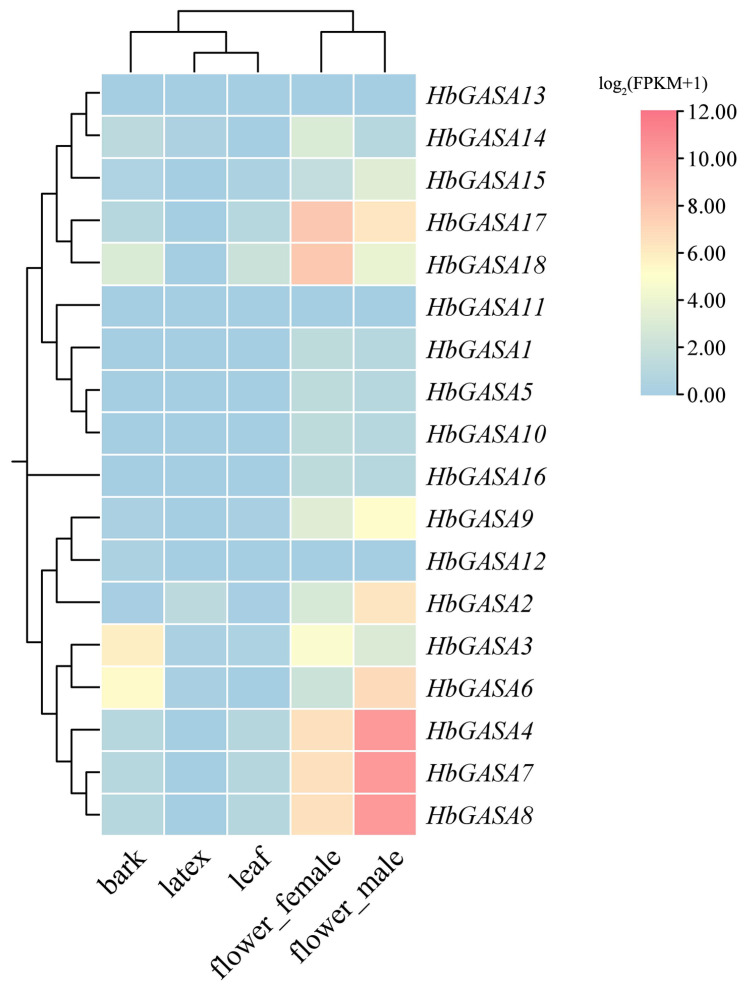
Expression profiles of *HbGASAs* based on transcriptome data.

**Figure 7 ijms-26-03454-f007:**
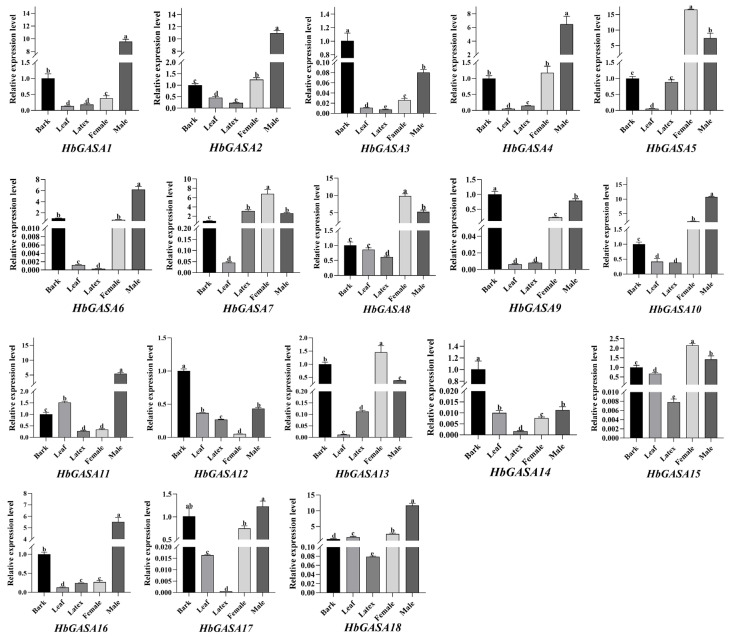
Tissue-specific expression profiles of *HbGASAs* detected by qRT-PCR. *HbYLS8* was used as an internal reference. Data were presented as the mean ± standard error (SE) of three biological replicates. Different lowercase letters indicated significant difference (*p* < 0.05).

**Figure 8 ijms-26-03454-f008:**
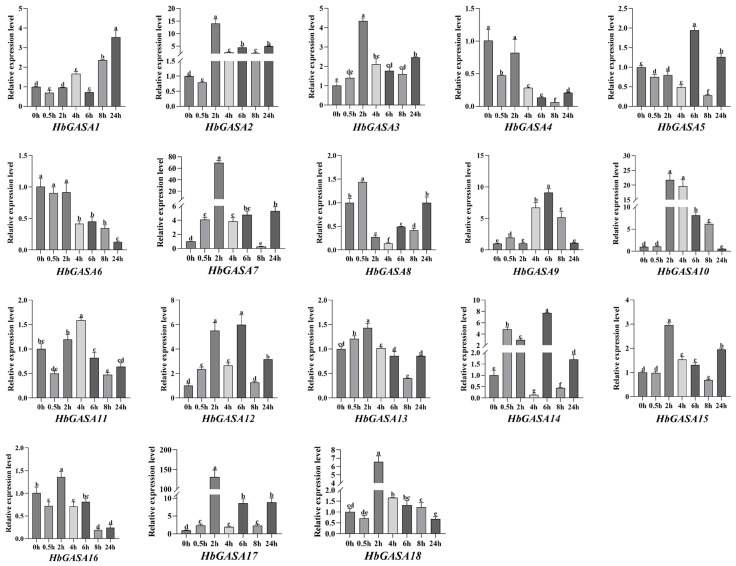
The expression patterns of *HbGASAs* detected by qRT-PCR under cold stress. *HbYLS8* was used as an internal reference. Data were presented as the mean ± standard error (SE) of three biological replicates. Different lowercase letters indicate significant difference (*p* < 0.05).

**Table 1 ijms-26-03454-t001:** Physiochemical characteristics of HbGASA proteins.

Gene Name	Gene ID	Protein (aa)	MW (Da)	pI	GRAVY	Instability Index	Aliphatic Index	Asp + Glu	Arg + Lys	Subcellular Localization
*HbGASA1*	LOC110646399	91	10,007.97	9.05	−0.071	35.77	74.07	5	14	Extracellular
*HbGASA2*	LOC110639973	88	9680.57	8.92	−0.006	27.25	57.61	6	14	Extracellular
*HbGASA3*	LOC110671420	94	10,413.46	9.12	−0.059	38.58	58.19	4	14	Extracellular
*HbGASA4*	LOC110645670	106	11,639.83	9.27	−0.176	37.78	61.60	2	14	Extracellular
*HbGASA5*	LOC110655875	100	10,764.64	7.46	0.184	64.30	82.90	7	8	Extracellular
*HbGASA6*	LOC110637234	112	12,424.58	9.45	−0.446	54.82	46.25	3	16	Extracellular
*HbGASA7*	LOC110665251	107	12,053.17	9.38	−0.407	45.44	45.61	2	15	Extracellular
*HbGASA8*	LOC110651389	107	12,116.24	9.42	−0.427	53.83	49.25	2	15	Extracellular
*HbGASA9*	LOC110667031	88	9709.73	9.33	−0.194	35.92	60.00	5	18	Chloroplast
*HbGASA10*	LOC110673766	104	11,066.93	8.92	−0.015	45.58	78.85	4	11	Extracellular
*HbGASA11*	LOC110646398	111	12,325.53	9.46	−0.154	56.89	73.78	4	18	Extracellular
*HbGASA12*	LOC110647740	98	10,406.24	9.16	−0.179	45.23	61.84	5	15	Extracellular
*HbGASA13*	LOC110643317	96	10,572.41	8.9	−0.328	50.72	63.85	8	15	Chloroplast
*HbGASA14*	LOC110632043	105	11,442.52	8.75	0.078	59.26	83.52	8	14	Extracellular
*HbGASA15*	LOC110660492	109	11,904.03	8.75	−0.083	46.10	79.54	9	15	Extracellular
*HbGASA16*	LOC110655855	106	11,499.55	9.01	−0.013	49.44	79.25	6	14	Extracellular
*HbGASA17*	LOC110667249	241	25,225.32	9.79	−0.226	62.00	66.64	9	32	Extracellular
*HbGASA18*	LOC110659225	253	26,546.86	9.98	−0.355	66.68	61.58	6	33	Extracellular

MW: molecular weight; pI: isoelectric point; GRAVY: grand average of hydropathicity; Asp + Glu: total number of negatively charged residues; Arg + Lys: total number of positively charged residues.

**Table 2 ijms-26-03454-t002:** Primers for qRT-PCR of *HbGASA* genes.

Primer Name	Forward Primer (5ʹ-3ʹ)	Reverse Primer (5ʹ-3ʹ)
*HbYLS8*	CCTCGTCGTCATCCGATTC	CAGGCACCTCAGTGATGTC
*HbGASA1*	AGCAAGGCATCAAGGCACAA	TTGGCATAGCAAGGGCAAAC
*HbGASA2*	GTGACTCCAAGTGTGGGGAG	ATAAGTTCCAGAGGGCACGC
*HbGASA3*	GCCTCAGGTGTACTTCCCTC	AGCACGAAACTTAGCACCCT
*HbGASA4*	ACCCAACACCACCTTGACAG	GTCTTGCTGCATCTCCTCGT
*HbGASA5*	TGCTTCCCTCCCACAACAAA	GCACAACACGTACCACATGC
*HbGASA6*	CAGCTTACAAGAAGCCGTGC	TTGTTGTAGCAAGGGCAGGA
*HbGASA7*	GTTATGGCATCGCATGGTCA	ATTGGGTCTTGCTGCACCTC
*HbGASA8*	GGCTAAATTTGTTGCTGCCTTCCT	GCCTTTGGTTGTCATAGTGGTG
*HbGASA9*	CAAGTCCCCTTACTGCACGA	AGAGGGCACACACTTGCATT
*HbGASA10*	ACCTTGCTCCAACCAATCCC	CAAGTCCCACATGCCCTCTT
*HbGASA11*	GCTCCTTCAGGTTTGTTTGGG	TCCTTGCATCTCCTTGAGCAT
*HbGASA12*	TCGTTCAGTTCACCAGGGCT	ATCCTGCATTCGCACACCTC
*HbGASA13*	TTGTGGAGGGTTGTGCAAGG	CACGCACTTGCACCTAAAGC
*HbGASA14*	TGAAGAGCAAAGCACCGAGTT	TGCAAACTACACCTCACCTTG
*HbGASA15*	GGCAGTCCGTTTGCTTCTTG	GAGCCTCCTGTTTCCACCTC
*HbGASA16*	ATGCGAGGGAAGGTGCAAAA	GGACAGAACTCAAGGTTGCC
*HbGASA17*	CCAAGGCTGTGTTGTTTCTGG	AGGAGTCTTGGCTGGTGGTA
*HbGASA18*	CCCAAATACTAGTCGCCCCA	GGAGTTGGTGCCTTGACTGG

## Data Availability

All data supporting the findings of this study are available within this article and the Appendix A published online.

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
