# Peer review of "Genome-Wide Identification and Expression Analysis of GASA Genes in Hevea brasiliensis Reveals Their Involvement in Response to Cold Stress"

_ijms, 2025, doi:10.3390/ijms26073454_

Round 1
Reviewer 1 Report
Comments and Suggestions for Authors
The manuscript entitled Genome-wide identification and expression analysis of GASA genes in Hevea brasiliensis reveals their involvement in response to cold stress. In this study, the authors have identified 18 HbGASA genes in the Hevea brasiliensis genome. They performed identification and characteristics of HbGASA genes, gene structure analysis, phylogenetic analysis, cis-acting elements analysis, and expression patterns analysis in five different tissues. Moreover, qRT-PCR examination was used to confirm the expression analysis of ten HbGASA genes in response under cold stress. The content of the original manuscript has a certain theoretical significance, but English writing mistakes need to be carefully polished and the logical relationship needs to be strengthened. However, there are some points that need improvement for its publication:
- The question of whether HbGASA is italicized as a transcription factor and whether PeAP2/ER is italicized as a gene throughout the full manuscript require unified proofreading.
- Why not use the maximum likelihood (ML) method to build the phylogenetic tree? The basis for the three exogenous species selected needs to be given.
- Choose the right software to draw Figure 3.
- Figures 1, 2, and 5 need to be adjusted for clarity and typography.
- The authors would like to focus on Figures 6 and 7. In Figure 6, HbGASA7, HbGASA8, HbGASA15, and HbGASA18 genes are expressed relatively low in Latex tissues, whereas the four genes are expressed higher in Figure 7. How can this be explained by the different expression trends of the same genes in the two figures? This question will have an impact on the conclusion of the whole paper and needs to be carefully considered and answered.
- However, this discussion is not sufficiently deep. The authors need to condense the scientific issues further and improve the quality of the manuscript.
The language of the paper requires major polish and refinement to improve overall fluency, coherence, and technical accuracy.
Author Response
Comments 1: The question of whether HbGASA is italicized as a transcription factor and whether PeAP2/ER is italicized as a gene throughout the full manuscript require unified proofreading.
Response 1: We have carefully checked the manuscript, and HbGASA, which is represented in italic and refers to a gene, has been proofread.
Comments 2: Why not use the maximum likelihood (ML) method to build the phylogenetic tree? The basis for the three exogenous species selected needs to be given.
Response 2: In general, if there is an appropriate molecular evolution model, the results obtained by constructing a phylogenetic tree using the Maximum Likelihood (ML) method are relatively good. For sequences of distantly related species, the Neighbor-Joining (NJ) method is usually employed. The basis for the four exogenous species, Arabidopsis, poplar, rice, and wheat, have be provided in part 4.6.
Comments 3: Choose the right software to draw Figure 3.
Response 3: We have redrawn Figure3.
Comments 4: Figures 1, 2, and 5 need to be adjusted for clarity and typography.
Response 4: We have updated these three figures.
Comments 5: The authors would like to focus on Figures 6 and 7. In Figure 6, HbGASA7, HbGASA8, HbGASA15, and HbGASA18 genes are expressed relatively low in Latex tissues, whereas the four genes are expressed higher in Figure 7. How can this be explained by the different expression trends of the same genes in the two figures? This question will have an impact on the conclusion of the whole paper and needs to be carefully considered and answered.
Response 5: The expression levels of gene showed some difference between two figures. HbGASA7, HbGASA8, HbGASA15, and HbGASA18 genes were expressed relatively low in latex according to RNA-seq data, and the four genes except HbGASA7 also displayed the lowest expression level in latex by qRT-PCR. The difference of HbGASA7 in expression patterns observed between qRT-PCR and RNA-Seq may be attributed to inherent differences between the two techniques: transcriptome sequencing quantifies all transcript variants of the gene, whereas qPCR may not capture all transcripts.
Comments 6: However, this discussion is not sufficiently deep. The authors need to condense the scientific issues further and improve the quality of the manuscript.
Response 6: We have already supplemented the discussion section.
Reviewer 2 Report
Comments and Suggestions for Authors
In this study, author used bioinformatics method to identify the 18 HbGASA genes in Hevea brasiliensis. HbGASA genes were involved in plant growth, development, and stress responses. Transcripts of the 18 HbGASA genes were detected in various tissues, and significant differences were observed in their expression levels. In addition, author found that some HbGASAs in response to cold stress play important roles. The methods and results are acceptable. There are some essential problems should be addressed by authors, which are listed below.
Major point
- This paper is just a simple analysis of bioinformation and qRT-PCR, lacking more relevant functional experiments. It is recommended that the author select 2-3 HbGASAs based on the experimental results in 2.5 and overexpress them in Arabidopsis thaliana or Nicotiana benthamiana to further confirm their subcellular localization and functions under cold stress.
Minor point
- The curves related to genes such as melting curves and gene expression curves should be added to the text.
- What is the basis for dividing GASA proteins into three subfamilies when constructing a phylogenetic tree, please list references.
- In 2.4, it is recommended to add qRT-PCR analysis of HbGASA at the root.
- The HbGASA4 gene also exhibits a high degree of tissue specificity. Why wasn’t it selected for qRT-PCR?
- HbGASA1, 5, 10 and 11 belong to the same family, but their expression signals were not detected by qRT-PCR. This is a rare case. Can you analyze the reason.
- Some Figures and Table should be modified.
In Figure 4, the ruler below the picture needs to add units “aa” and “bp”.
In Figure 5A, all zeros are cleared.
In Figure 6, please indicate the calculation method and add it to the right side of the scale, for example, Log2(FPMK+1).
Comments on the Quality of English LanguageSee comments
Author Response
Major point
Comments 1: This paper is just a simple analysis of bioinformation and qRT-PCR, lacking more relevant functional experiments. It is recommended that the author select 2-3 HbGASAs based on the experimental results in 2.5 and overexpress them in Arabidopsis thaliana or Nicotiana benthamiana to further confirm their subcellular localization and functions under cold stress.
Response 1: The reviewer's suggestions are highly valuable. In the current study, we concentrated on the genome-wide identification of HbGASAs, and identified preliminary evidence suggesting their potential involvement in cold stress responses. In subsequent studies, more in-depth experiments need to be carried out to comprehensively elucidate the functions of HbGASAs under cold stress.
Minor point
Comments 1: The curves related to genes such as melting curves and gene expression curves should be added to the text.
Response 1: The relevant melting curves have been added in part 2.5.
Comments 2: What is the basis for dividing GASA proteins into three subfamilies when constructing a phylogenetic tree, please list references.
Response 2: The relevant references have been listed in the Discussion.
Comments 3: In 2.4, it is recommended to add qRT-PCR analysis of HbGASA at the root.
Response 3: In most plant studies, target gene expression is typically analyzed in both shoot and root tissues. However, rubber tree seedlings are commonly propagated via grafting, where the root system originates from the rootstock while the aerial tissues derive from the scion. Given this grafting practice, the expression levels of HbGASAs were not determined in root tissues in this study. This decision was made to avoid potential confounding effects from genetic differences between the rootstock and scion, which could complicate interpretation of expression patterns specific to the scion-derived shoot tissues.
Comments 4: The HbGASA4 gene also exhibits a high degree of tissue specificity. Why wasn’t it selected for qRT-PCR?
Response 4: All 18 HbGASA genes have been investigated by qRT-PCR.
Comments 5: HbGASA1, 5, 10 and 11 belong to the same family, but their expression signals were not detected by qRT-PCR. This is a rare case. Can you analyze the reason.
Response 5: All 18 HbGASA genes have been investigated by qRT-PCR.
Comments 6: Some Figures and Table should be modified.
In Figure 4, the ruler below the picture needs to add units “aa” and “bp”.
In Figure 5A, all zeros are cleared.
In Figure 6, please indicate the calculation method and add it to the right side of the scale, for example, Log2(FPMK+1).
Response 6: The corresponding charts have been revised.
Reviewer 3 Report
Comments and Suggestions for Authors
The authors performed identification of GASA gene family through genome-wide analysis of physicochemical characteristics, phylogenetic analysis, cis-acting analysis, chromosome location and expression analysis. And authors showed HbGASA18 gene was isolated and its expression and function were confirmed. Overall, the study sounds well about the findings. However, I have some suggestions to improve the presentation of this manuscript.
- In table 1, add Instability and aliphatic index.
- In table 1, add total number of negatively charged residues ((Asp + Glu) and total number of positively charged residues (Arg + Lys).
- To characterize this gene family, it is necessary to show the expression of all HbGASA 18 genes, not just some of the genes, as shown in Fig 7 and 8.
- In Fig 7 and Fig 8, statistical analysis of qPCR results is required.
- In results, authors should provide collinearity and synteny analysis.
- The cis of cis-acting should be italicized.
Author Response
Comments 1:
- In table 1, add Instability and aliphatic index.
- In table 1, add total number of negatively charged residues (Asp + Glu) and total number of positively charged residues (Arg + Lys).
Response 1: The relevant data have been added to Table 1.
Comments 2:
- To characterize this gene family, it is necessary to show the expression of all HbGASA 18 genes, not just some of the genes, as shown in Fig 7 and 8.
- In Fig 7 and Fig 8, statistical analysis of qPCR results is required.
Response 2: All 18 HbGASA genes have been investigated by qRT-PCR, and statistical analysis of qRT-PCR results were performed.
Comments 3:
- In results, authors should provide collinearity and synteny analysis.
Response 3: Collinearity analysis has been added.
Comments 4:
- The cis of cis-acting should be italicized.
Response 4: We have revised it in the article.
Round 2
Reviewer 1 Report
Comments and Suggestions for Authors
The manuscript entitled Genome-wide identification and expression analysis of GASA genes in Hevea brasiliensis reveals their involvement in response to cold stress. The changes are better in this version, but there are still some minor points that need to be addressed. I believe that this manuscript is now suitable for publication.
- This manuscript contains mistakes in the English grammar. I recommend that the authors use the help of a native English speaker or send the manuscript to an English Editing Service that proofreads scientific writing.
- It is necessary to include a “Statistical analysis” subsection in the Materials and Methods.
- More information on statistical analysis should be included in each Figure where statistical analysis has been applied. For example, see Figures 9 and 10. What is provided here, standard error or standard deviation? It should also be mentioned that data are the means. Mention the statistical test used to determine statistical significance.
- Please increase the image figures and enhance the typography so that readers can see the content clearly.
This manuscript contains mistakes in English grammar. I recommend that the authors should use some help of a native English speaker or send the manuscript to an English Editing Service that proofreads scientific writing.
Author Response
Comments 1: This manuscript contains mistakes in the English grammar. I recommend that the authors use the help of a native English speaker or send the manuscript to an English Editing Service that proofreads scientific writing.
Response: With sincere gratitude for the reviewer’s constructive feedback, the manuscript has been meticulously revised by a native English-speaking academic expert to ensure linguistic precision and adherence to journal standards.
Comments 2: It is necessary to include a “Statistical analysis” subsection in the Materials and Methods.
Response: Statistical analysis has been incorporated into Section 4.9.
Comments 3: More information on statistical analysis should be included in each Figure where statistical analysis has been applied. For example, see Figures 9 and 10. What is provided here, standard error or standard deviation? It should also be mentioned that data are the means. Mention the statistical test used to determine statistical significance.
Response: Figure legends for Figures 9 and 10 have been revised to enhance clarity and consistency in response to reviewer feedback.
Comments 4: Please increase the image figures and enhance the typography so that readers can see the content clearly.
Response: All figures have been updated.
Reviewer 2 Report
Comments and Suggestions for Authors
Good work for the revision.
Author Response
Comments 1: Good work for the revision.
Response 1: We sincerely thank the reviewer for the hard work and dedication
Reviewer 3 Report
Comments and Suggestions for Authors
The authors made the corrections and took into the considerations all my comments.
Thank you.
Author Response
Comments 1: The authors made the corrections and took into the considerations all my comments.
Response 1: We deeply appreciate the reviewer's insightful comments and thorough evaluation of our manuscript.